# Teacher Morale and Mental Health at the Conclusion of the COVID-19 Pandemic

**Tim Pressley** [1,*] , **David T. Marshall** [2] , **Savanna M. Love** [3] **and Natalie M. Neugebauer** [2]

1   Department of Psychology, Christopher Newport University, Newport News, VA 23606, USA
2   Department of Educational Foundations, Leadership, and Technology, Auburn University, Auburn, AL 36849, USA; dtm0023@auburn.edu (D.T.M.)
3   Department of Education, Randolph-Macon College, Ashland, VA 23005, USA; savannalove@rmc.edu
*   Correspondence: timothy.pressley@cnu.edu

**Abstract:** As teachers entered the 2021–2022 school year, schools tasked teachers with the challenge of closing the learning loss gap, maintaining the same standards as a typical school year, implementing COVID-19 safety protocols and teaching to grade-level standards. The current study used the theoretical framework of teacher demoralization to explore teachers' experiences of morale and mental health at the end of the 2021–2022 school year. The participants included 830 PK-12 individual teachers from across the United States who completed an online survey. The data were analysed using Pearson's correlation and the hierarchical multiple regression model to determine the predictor variables. The results showed significant predictors for both teacher morale and mental health. The implications focus on steps that school leaders can take to support teachers' morale and mental health.

**Keywords:** teacher morale; teacher mental health; COVID-19





## 1. Introduction

Teaching is a hard job. Even before the COVID-19 pandemic, teacher attrition had been a long-standing problem, with almost half of teachers leaving within the first five years [1,2]. Teaching did not get any easier during the COVID-19 pandemic as teachers became frontline workers providing instruction to students in many different formats. Throughout the pandemic, teachers took on new and unprecedented challenges to provide instruction to students [3,4]. Research also found that teachers had high stress levels and felt overworked [5–8]. For example, Pressley [9] found that teachers had higher stress levels due to the expectations of teaching multiple formats (in-person and HyFlex students) and the lack of guidance from the administration to overcome the new challenges of teaching during a pandemic. A silver lining of the pandemic for some teachers and school leaders during this time was that it forced educators to connect with parents and students more intentionally—especially when school was remote. However, not all interactions with families were positive. The negative feedback and lack of support from some parents also took a toll on teacher stress [10,11].

Other studies conducted throughout the pandemic found increased teacher anxiety and depression [12–14]. This had an overall impact on teacher well-being and led to lower morale and life-work balance by the end of the 2020–2021 school year [15]. However, some teachers found ways to limit the mental strain. For example, Walter and Fox [14] found that teachers with empathetic leadership, good working teams, and increased resources had a better sense of well-being than other teachers.

The impact of the pandemic went beyond teacher stress and burnout. Teacher self-efficacy also decreased as teachers navigated the ever-changing requirements and expectations [14,16]. Thus, many teachers felt they were underperforming pedagogically due to the new requirements and expectations during this time. In addition, as teachers tried to adjust to new teaching approaches and feel confident in their teaching, they also

faced increased anxieties about teaching during a pandemic with very little input on return to learning plans [14,17]. Overall, society asked a lot from teachers during the first year and a half of the pandemic, which significantly impacted their well-being.

As teachers entered the 2021–2022 school year, schools across the United States had a variety of requirements in place for teachers to enforce to keep students safe and faced continued uncertainty as the COVID-19 pandemic progressed [14]. These safety protocols would change throughout the year, leading to inconsistencies across states and schools with teachers having very little say in policy changes. On top of the safety protocols, schools were pushed to return to more normalcy and to address the learning loss that occurred during the pandemic [18]. This included the return of mandated state standardized tests, new curriculum initiatives, and lofty teacher expectations for the 2021–2022 school year. Enforcing safety protocols and additional pressure to raise student achievement were just two of the requirements and expectations put on teachers' shoulders. Survey research conducted at the conclusion of the 2021–2022 school year found that 29.5% of teachers reported having low levels of morale [15]. As such, it is important to understand the toll that teaching during the pandemic had on teachers' morale and mental health. Thus, the aim of this study was to improve teaching conditions by exploring teachers' experiences of the COVID-19 pandemic and its impact on their morale and mental health by asking the following questions.

1. What was the state of teacher morale and mental health at the end of the 2021–2022 school year?
2. What variables predicted teacher morale at the end of the 2021–2022 school year?
3. What variables predicted teachers' mental health at the end of the 2021–2022 school year?

## 2. Literature Review

Demoralization occurs when teachers believe they are "violating basic moral expectations that educators should embody: do not harm students, support student learning, engage in behavior becoming of a professional" [19] (p. 43). Teachers faced new demoralization issues throughout the pandemic that challenged their morale [20]. These challenges included instructional requirements that did not meet students' learning needs, limited parental support [21,22], and limited mental health resources [4,23]. These aspects have led teachers to have a lower teacher self-efficacy [24,25], higher levels of burnout [11,13,26], and higher levels of stress and anxiety [8,27], with risks to overall teacher well-being.

Demoralization specifically focuses on "consistent and persistent frustrations in accessing the moral rewards of teaching" and leads teachers to "feel they no longer can do good work or teach 'right'" [28] and thus feel they must leave the classroom. Demoralization is especially important in times of change [28]. Though demoralization may encompass aspects of teacher burnout, such as depression or shame, demoralized teachers will have continued frustration in providing the best instruction to their students [28]. This often occurs when teachers face a change they perceive no longer allows them to provide good teaching [19]. Thus, new expectations and requirements from school leaders can often lead to teacher demoralization [19,28].

Morale has been linked to mental health in a variety of contexts. In discussing teaching conditions, Edwards [29] argues that teachers must be secure in a position where "personal integrity is maintained, and group harmony is fostered" to attain high morale. He goes on to state that societal and school factors make this almost impossible to achieve, and as a result, "the morale of teachers is at a low ebb and their total mental health adversely affected" (p. 18). In a medical context, Mojoyinola [30] connected job stress to work morale, finding that job stress significantly impacted nurses' health and mental health. Additionally, morale has been linked to mental health in military contexts [31,32], with findings indicating that higher morale levels lead to lower instances of PTSD. More recently, Kendrick [33] found that low morale had a negative impact on the mental health of academic librarians, citing that "demoralized academic librarians experience depression and related conditions, including PTSD" (p. 873).

### 2.1. Teacher Morale

Teacher morale is defined as the degree to which a person's needs are satisfied and the person's perception of how the job situation brought that state of satisfaction of the worker to fruition [34]. Even before the COVID-19 pandemic, teacher morale has been shown to positively correlate with student achievement [35,36]. In addition, low teacher morale negatively impacts teacher productivity and results in a detachment from students [37,38]. Furthermore, factors such as motivation, effort, and job satisfaction have been linked to teacher morale [39–41]. Dunn [42] discusses how Coughlan's 1970 work, in which he found 13 factors that affected teacher morale, is still relevant today. These factors were organized into four categories: administrative operations, working relationships, school effectiveness, and career fulfillment [43]. More recent literature on teacher morale has discussed the impact of intrinsic and extrinsic factors on job satisfaction [40]. Intrinsic factors include job security, ability utilization, and social service, and extrinsic factors include compensation, authority, school policies, the opportunity for advancement, recognition, and power distribution. Compensation was found to be a significant factor for teacher morale in a number of studies [38]. The literature has also demonstrated a relationship between teacher autonomy and their morale [44]. Teachers who feel they can make decisions that best meet the needs of their students and who are trusted to use their unstructured professional time also report higher levels of morale. Teacher morale has continued to be an important factor during the COVID-19 pandemic. Research has found a relationship between morale and an intention to remain in the profession [20]. With research before and during the COVID-19 pandemic indicating the importance of teacher morale, researchers, school leaders, and teachers should continue to develop a better understanding of teacher morale as schools move beyond the COVID-19 pandemic.

### 2.2. Teacher Mental Health

Previous studies have explored the impact of the COVID-19 pandemic on students' mental health and well-being across different educational settings [45]. However, researchers have only recently started investigating teachers' mental health and well-being, as researchers have associated declining teachers' mental health and well-being during COVID-19 with many teachers leaving the teaching profession [46]. For example, during the fall of 2020, teachers' mental health and well-being drastically declined due to a variety of additional demands at work, such as an increased workload, multiple roles (i.e., covering colleagues' classes), and a sense of uncertainty, as well as a lack of resources at work, including social support, work autonomy, and coping skills [47].

In addition, the challenges of online teaching, including the lack of connection to students, have been shown to negatively impact teacher well-being and mental health [48,49]. The lack of connection to students that many teachers experienced and the inability to build meaningful relationships with their students affected teachers' mental health and posed a significant barrier to their personal and professional development [50]. Kush and colleagues [51] found a striking difference in teachers' mental health based on teaching format. Kush et al. surveyed 135,488 teachers during the 2020–2021 school year and found that teachers who taught in person were significantly less likely to report feelings of depression or isolation compared to peers teaching remotely. Teachers also reported high levels of stress, burnout, and job dissatisfaction due to the challenges they faced online teaching during COVID-19 [14,52].

Similar to the online environment, teachers who taught in a HyFlex environment also saw an impact on their mental health [47,53]. Kim and colleagues [47] surveyed teachers across three-time points in the 2020–2021 school year and found that teachers' stress and burnout reached an all-time high in November 2020 when HyFlex teaching was introduced in schools in the United States. Teachers' narratives reflected that HyFlex teaching was particularly challenging, as they were never trained to teach in such a way. Similarly, Pressley et al. [8] found that HyFlex teachers had the greatest increase in anxiety during the fall of 2020 when teachers returned back to school after the United States shut down schools in the spring of 2020.

While instructional methods have varied and changed throughout the pandemic, teacher morale and mental health remain a concern. With the increase in teacher attrition and more being asked of teachers, having a better understanding of teachers' morale and mental health is critical for school leaders. Providing environments that support teachers could be important ways to keep teachers in the classroom [46].

## 3. Method

This exploratory study focused on understanding potential variables related to teacher morale and mental health at the end of the 2021–2022 school year. Specifically, we used a survey design to reach a large number of teachers across the United States.

### 3.1. Procedures and Sampling

To understand teachers' experiences of the impact of the COVID-19 pandemic on their morale and mental health, the researchers used convenience and snowball sampling from 4–18 May 2022. The researchers selected this time period as a majority of teachers were within the last few weeks of the school year. This would also allow teachers to reflect back on the year, which for many included changes in COVID-19 policies throughout the school year. To be eligible to participate in the research, individuals had to be currently employed as PK-12 teachers in the United States at the time of the survey administration. This research received IRB approval prior to the start of research activities. Researchers distributed an anonymized link to the survey to personal networks and posted the link on social media sites, including Facebook, Twitter, and Reddit. Researchers also emailed the survey to recent graduates of a teacher preparation program. Using social media platforms allowed for an extensive sample of teachers from across the United States to participate in the current study. Finally, researchers encouraged teachers to share the survey with other teachers if they were so inclined.

### 3.2. Participants

The current sample included 830 PK-12 individual teachers from across the United States ($N$ = 49) states; however, if a participant had missing data, they were not included in the final analysis. Participants were 40 years of age ($M$ = 40.41, $SD$ = 10.38), had almost 13 years of teaching experience ($M$ = 13.26, $SD$ = 8.97), and were overwhelmingly White (86.8%), female (87.8%), and taught at the elementary level (PK-5; $n$ = 339). Just over half of the participants (61.4%) taught in a Title 1 school, and more than four in five (82.8%) taught in a traditional public school. Almost half of the participants (48.1%) taught in a suburban school; about a quarter taught in an urban setting (25.3%), and the rest taught in either a rural or small-town setting. See Table 1 for descriptive statistics of the study sample.

**Table 1.** Descriptive statistics for the study's sample.

| Variable | N | % | M | SD |
|---|---|---|---|---|
| Age | | | 40.41 | 10.38 |
| Years of Teaching Experience | | | 13.26 | 8.97 |
| *Gender* | | | | |
|   Female | 729 | 87.8 | | |
|   Male | 91 | 11.0 | | |
|   Other | 9 | 1.1 | | |
| *Race/Ethnicity* | | | | |
|   African American/Black | 31 | 3.7 | | |
|   Asian American | 17 | 2.1 | | |
|   Hispanic | 28 | 3.4 | | |
|   Jewish [a] | 2 | <0.1 | | |
|   Native American/Alaskan | 1 | <0.1 | | |
|   Pacific Islander/Hawaiian | 5 | 0.6 | | |
|   White/Caucasian | 721 | 87.1 | | |
|   More than one race | 21 | 2.5 | | |

**Table 1.** *Cont.*

| Variable | N | % | M | SD |
|---|---|---|---|---|
| Title 1 | 508 | 61.4 | | |
| Special Education Teacher | 81 | 9.8 | | |
| *School Type* | | | | |
| Traditional Public School | 686 | 82.8 | | |
| Magnet School | 34 | 4.1 | | |
| Charter School | 57 | 6.9 | | |
| Private School | 52 | 6.3 | | |
| *Geographic Location* | | | | |
| Rural | 120 | 14.5 | | |
| Small Town | 100 | 12.0 | | |
| Suburban | 399 | 48.1 | | |
| Urban | 210 | 25.3 | | |

Note: [a] Self-described.

### 3.3. Survey Development

Several factors were considered when developing the survey used to answer this study's research questions. Research we had conducted during the pandemic prior to this study suggested that teachers' workloads, stress levels, and burnout were elevated during the crisis [6,11,15]. Additional studies echoed similar findings [7,13,25,54,55]. As such, we opted to measure several constructs with a single item to reduce the burden placed on teachers to complete the survey. We believed this design was appropriate in order to capture teachers' experiences during an important moment during the COVID-19 pandemic.

#### 3.3.1. Teacher Morale, Teacher Mental Health, and Student Mental Health

To understand teacher morale, teacher mental health, and student mental health, participants responded to a single item on a 0- (*very bad*) to 10-point (*very good*) scale. Teacher morale was measured similar to how Senechal et al. [56] measured the construct asking participants to respond to the question, "In terms of how you currently feel, rate your morale as a teacher?" We similarly asked participants to respond to items about their mental health and their students mental health with the questions, " In terms of how you currently feel, rate your mental health?" and "How would you currently rate your students' mental health?"

#### 3.3.2. Administrative Support

The survey included a scale measuring administration support [57]. Teachers were presented with questions about their perspectives on a 6-point scale ranging from (1) *strongly disagree* to (6) *strongly agree*. Example questions included "I believe that my efforts in the classroom are unappreciated by the administrators". In this study, the administrative support scale yielded strong reliability ($\alpha = 0.91$).

#### 3.3.3. Teacher Autonomy

Teacher autonomy was measured with the Teacher Leadership and Autonomy Scale [58]. Similar to the administrative support scale, teachers were asked to respond in terms of the extent to which they agreed with each item on a 6-point scale. Example questions included "I control how I use my scheduled classroom time". In this study, the teacher autonomy scale yielded strong reliability ($\alpha = 0.85$).

#### 3.3.4. Student Work Completion and Parent Support

To measure student work completion, a single 6-point Likert scale item was included that asked teachers about the extent they agreed with the statement, "Most of my students complete all assigned work". Similarly, to measure parent support, a single 6-point Likert scale item was included that asked participants about the extent to which they agreed with the statement, "My students' parents have been supportive of my teaching".

*3.4. Data Analysis*

　　Two separate models were tested to better understand predictors of teacher morale and teacher mental health. Separate models were tested, one with teacher morale as the dependent variable and one with teacher mental health as the dependent variable. Pearson's correlations were conducted to determine the association between variables. Each of the predictor variables included in the models was significantly correlated with each other. Additionally, the researchers conducted a hierarchical multiple regression model to test predictor variables, controlling for ethnicity, gender, and years of teaching experience. Lastly, the researcher screened the data to check that the assumptions for multiple regression analyses were met. No issues were found with singularity, multicollinearity, the dependence of errors, normality, linearity, or homoscedasticity of residuals [59]. All analyses were conducted in SPSS version 28.

**4. Results**

　　Teachers were asked to rate their experiences of morale and mental health on a 10-point scale. As such, the average teacher morale ($M = 3.76$) and teacher mental health ($M = 3.95$) suggested that teachers had low morale and mental health at the end of the 2021–2022 school year.

　　A Pearson's correlation was conducted to determine possible predictors of teachers' morale and teachers' mental health at the end of the 2021–2022 school year. For teachers' morale, the results suggested positive associations with teachers' mental health ($r = 0.64$), administrative support ($r = 0.44$), teacher autonomy ($r = 0.43$), student mental health ($r = 0.40$), student work completion ($r = 0.34$), and parent support ($r = 0.30$).

　　For teachers' mental health, the results suggested positive associations with teachers' morale ($r = 0.64$), administrative support ($r = 0.37$), teacher autonomy ($r = 0.37$), student mental health ($r = 0.40$), student work completion ($r = 0.30$), parent support ($r = 0.27$), and teaching experience ($r = 0.10$; see Table 2 for correlations).

**Table 2.** Correlation matrix of variables included in models.

|  | I | II | III | IV | V | VI | VII |
|---|---|---|---|---|---|---|---|
| I. Teacher Morale | 1 | | | | | | |
| II. Teacher Mental Health | 0.638 ** | 1 | | | | | |
| III. Administrative Support | 0.438 ** | 0.371 ** | 0.405 ** | 1 | | | |
| IV. Teacher Autonomy | 0.429 ** | 0.371 ** | 0.394 ** | 0.573 ** | 1 | | |
| V. Student Mental Health | 0.397 ** | 0.396 ** | 0.275 ** | 0.198 ** | 0.162 ** | 1 | |
| VI. Student Work Completion | 0.340 ** | 0.297 ** | 0.373 ** | 0.265 ** | 0.229 ** | 0.373 ** | 1 |
| VII. Parental Support | 0.296 ** | 0.268 ** | 0.380 ** | 0.273 ** | 0.324 ** | 0.263 ** | 0.410 ** |

Note: ** $p < 0.01$.

　　The same set of predictor variables was used for both the teachers' morale and mental health regression models. This allowed for us to compare predictors across models. Because of missing data, each model only included participants who completed all of the measures for each variable included in the model.

*4.1. Teachers' Morale*

　　After controlling for ethnicity, years of experience, gender, Title 1 status, and urbanicity, the regression indicated that the model explained 34% ($R^2 = 0.340$) of the variance, and the model was significant $F(11, 745) = 37.724$, $p < 0.001$. Specifically, six variables remained as significant predictors for teachers' morale, including non-White race/ethnicity ($\beta = 0.411$, $p = 0.026$), Title I school status ($\beta = -0.391$, $p = 0.011$), administrative support ($\beta = 0.065$, $p < 0.001$), teacher autonomy ($\beta = 0.564$, $p < 0.001$), student mental health ($\beta = 0.213$, $p = 0.001$), and student work completion ($\beta = 0.172$, $p = 0.002$).

To explore further differences in teachers' morale, we also ran an independent-samples *t*-test to compare the teachers' morale scores based on race/ethnicity. A Levene's test found that the assumption of homogeneity of variances was met ($p = 0.527$), which was important given the difference in sample size between White and non-White teachers. Teachers who identified as White had a slightly lower morale level ($M = 3.67$, $SD = 2.29$), compared to teachers who identified as non-White ($M = 4.14$, $SD = 2.42$). This difference approached statistical significance, with a small measure of practical significance $t(776)= 1.91$, $p = 0.051$, $d = 0.21$. We also operationalized teacher morale similar to Senechal and colleagues [56], coding a response of 0–3 as low morale, a response of 4–5 as medium-low morale, a response of 6–7 as medium-high morale, and a response of 8–10 as high morale. More than half of the sample (53.6%) reported having low morale. A Chi-Square goodness-of-fit test found that participants reported low morale significantly more often than any other level of morale ($X^2$(3 d.f.) = 394.410, $p < 0.001$). See Table 3 for beta coefficients and standard errors for the teacher morale regression models.

**Table 3.** Regression findings for teacher morale models.

| | Model I | Model II |
|---|---|---|
| **Variable** | **β(SE)** | **β(SE)** |
| Teaching Experience | 0.011 (0.009) | −0.008 (0.008) |
| Female | 0.258 (0.259) | 0.219 (0.213) |
| Non-White | **0.566 (0.254) *** | **0.411 (0.206) *** |
| Title 1 | −0.091 (0.176) | **−0.367 (0.145) *** |
| Rural | 0.250 (0.246) | 0.137 (0.202) |
| Urban | **−0.447 (0.204) *** | −0.178 (0.167) |
| Administrative Support | | **0.065 (0.011) **** |
| Teacher Autonomy | | **0.564 (0.091) **** |
| Student Mental Health | | **0.312 (0.037) **** |
| Student Work Completion | | **0.172 (0.055) *** |
| Parental Support | | 0.114 (0.060) + |
| | | |
| N | 757 | 757 |
| F | 2.238 * | 37.724 *** |
| F Δ | 2.238 * | 78.832 *** |
| $R^2$ | 0.018 | 0.358 |
| $R^2$Δ | 0.018 | 0.340 |

Note: *** $p < 0.001$; ** $p < 0.01$; * $p < 0.05$; + $p < 0.10$.

### 4.2. Teacher Mental Health

After controlling for ethnicity, years of experience, gender, Title 1 status, and urbanicity, the regression indicated that the model explained 30% ($R^2 = 0.304$) of the variance, and the model was significant $F(11, 754) = 29.892$, $p < 0.001$. Specifically, six variables remained as significant predictors for teachers' mental health, including Title I school status (β = −0.593, $p = 0.020$), teaching in an urban school (β = −0.593, $p < 0.001$), administrative support (β = 0.040, $p < 0.001$), teacher autonomy (β = 0.480, $p < 0.001$), student mental health (β = 0.306, $p < 0.001$), and student work completion (β = 0.112, $p = 0.037$). A follow-up item asked participants if they had seen a mental health counselor during the COVID-19 pandemic, and 30.92% responded affirmatively. See Table 4 for beta coefficients and standard errors for the teacher mental health regression models.

**Table 4.** Regression findings for teacher mental health models.

|  | Model I | Model II |
|---|---|---|
| **Variable** | **β (SE)** | **β (SE)** |
| Teaching Experience | **0.027 (0.009) \*\*** | 0.012 (0.007) |
| Female | 0.011 (0.238) | −0.040 (0.207) |
| Non-White | 0.302 (0.235) | 0.161 (0.201) |
| Title 1 | −0.111 (0.162) | **−0.327 (0.140) \*** |
| Rural | −0.041 (0.227) | −0.111 (0.195) |
| Urban | **−0.831 (0.187) \*\*\*** | **−0.593 (0.162) \*\*\*** |
| Administrative Support |  | **0.040 (0.011) \*\*\*** |
| Teacher Autonomy |  | **0.480 (0.088) \*\*\*** |
| Student Mental Health |  | **0.306 (0.036) \*\*\*** |
| Student Work Completion |  | **0.112 (0.054) \*** |
| Parental Support |  | 0.106 (0.058) + |
| N | 766 | 766 |
| F | 5.216 \*\*\* | 29.892 \*\*\* |
| F Δ | 5.216 \*\*\* | 57.186 \*\*\* |
| $R^2$ | 0.040 | 0.304 |
| $R^2$Δ | 0.040 | 0.264 |

Note: \*\*\* $p < 0.001$; \*\* $p < 0.01$; \* $p < 0.05$; + $p < 0.10$.

## 5. Discussion

The results of this exploratory study suggest that teachers' experienced low levels of morale and mental health at the conclusion of the 2021–2022 school year. A similar survey was conducted at the conclusion of the 2020–2021 school year [15]. In that study, the researchers similarly operationalized teacher morale [56] and found that 29.5% of teachers reported low levels of morale, compared to 53.6% in May 2022—an 82% increase over the span of a single year. The current study also found that teachers reported low levels of mental health, and 30.9% reported having received mental health counseling during the pandemic—much greater than the Center for Disease Control and Prevention's most recent national estimation that 20.3% of Americans see a counselor [60]. To be clear, we do not assume that seeing a counselor is necessarily negative; however, we find these figures to be concerning, and follow-up tests found that participants in our sample who saw a counselor reported lower levels of mental health than those who did not. The current study's descriptive statistics align with previous studies conducted throughout the pandemic: teaching during COVID-19 has taken a toll on teachers mentally (e.g., Pellerone, [16]) and physically (e.g., Kotowski et al. [61]).

Across both models, the data suggest significant predictors explaining close to a third of the variance of each dependent variable. It is important to note that for both teachers' morale and mental health, the variables of teaching at a Title I school, teaching at an urban school, administrative support, student mental health, and student work completion were all significant predictors. One positive to note from the research is that teachers who identified as a minority had slightly higher morale compared to teachers who identified as white within the sample. Though the difference was small, it suggests that minority teachers may have felt more support or teacher autonomy to support students compared to their white peers and, thus, may feel less demoralized. These findings align with other studies completed during the pandemic, which suggest that minority teachers may have fared better mentally during the pandemic [48]. Baker et al. suggest that this may be due to underreporting or may imply resilience within this subgroup [48]. Future research should continue to explore this aspect in more detail to learn more about these results.

As the educational system has shifted and evolved throughout the pandemic, there has been an increase in the number of demoralized teachers [19]. The current findings suggest that factors such as the school environment, teacher autonomy, teacher concerns about student mental health and motivation, and administrative support influence teachers' morale and mental health. The lack of teachers' autonomy and administrative support may

suggest that teachers do not always feel they can provide the best support to students as they see fit [28]. Low teacher morale and mental health may lead to teachers feeling more depressed or ashamed of their inability to support students in the best possible way and may lead to higher teacher attrition rates. In addition, teachers may feel more pressure from administrators to raise student achievement. Students completed less work and learned less overall than prior to the pandemic, and this was exacerbated for students who attended schools that spent more time engaging in remote learning [18,62,63]. This left many teachers feeling low morale as helping students catch up and learn new material became increasingly difficult [28].

The current study's findings align with previous work on teacher well-being and the role of administrative support [12–14]. Specifically, the low teacher morale is unsurprising as Marshall et al. [15] found teachers had lower morale at the end of the 2020–2021 school year. What is concerning is that the current findings suggest the low morale has continued through another school year. With the potential impact on student achievement [36] and relationships with students [37], it is important for school leaders to take immediate action. As supported by the current predictor variables, school leaders might consider intrinsic and extrinsic [40], administrative operations, working relationships, and school effectiveness categories [42] as potential first steps for supporting teacher morale.

Similar to previous literature, the current results suggest low teacher mental health [46]. Specifically, administrative support and the resources available to teachers play a key role in teacher mental health [47]. Beyond the support teachers receive, students also play a key role in teacher mental health as the current study found student mental health and work completion as a significant predictors. Though previous studies have not explored these aspects, other studies have found the relationships teachers build with their students play a significant role in teacher mental health [50,51]. This was especially true for teachers in an online environment [14,52]. Therefore, school leaders might consider finding resources for teachers to support students within their class(es) and provide resources for teachers, which may also increase job satisfaction and decrease feelings of burnout and demoralization [46,47].

### 5.1. Implications

Supporting teacher morale and mental health is always important, but this is especially true after a time of crisis, such as the COVID-19 pandemic. As schools continue to push back to a pre-pandemic approach to learning, school leaders still need to be aware of and find ways to support teacher morale and mental health. Support is especially important for teachers in Title I or Urban school settings. This may begin with providing teachers school-level support for their instruction and mental health. A demoralized teacher does not feel that they are supporting their students in the best possible way. School administrators can start by communicating with teachers about their needs and finding ways to support teachers and students. Teachers spend the most time with students and may be able to gauge the most appropriate ways to support student needs rather than a program pushed from the top-down. School leaders may consider looking into mental health programs that can be brought into the classroom to support teachers' and students' mental health. This may also lead to teachers feeling more autonomy in their classrooms and feeling less demoralized as they can support their students and their own needs. Schools can also support teachers' mental health by providing telehealth resources to access professional counselors or providing mental health days to teachers throughout the school year.

Another way school leaders may support teacher morale and mental health is by encouraging teachers to support each other at the grade/subject level and the school level. This may include providing protected planning time to meet as a grade level or giving teachers an opportunity to voice perspectives during faculty meetings or through surveys [39]. School leaders can use this feedback to provide a supportive environment that would benefit the teachers. Encouraging a supportive environment may help teachers feel that they are not alone and build more of a team environment rather than an individual one [14].

Additionally, teachers need support from their administrators. This support may come through resources but also flexibility and understanding. Throughout the pandemic, teachers have shared feelings of being overworked, limited flexibility in instructional approaches, and increased expectations as schools return to normalcy [14]. School leaders must provide flexibility and set realistic expectations for teachers, understanding that students are behind and trusting teachers to make sound decisions to support student learning [10,14]. School leaders can provide flexibility when available by providing a substitute teacher for an extra planning period, provide mental health days by allowing teachers to work from home on days students are not in the building, or limit the requirements put on teachers, such as extra paperwork or implementing new curriculum materials.

Finally, schools need to support teachers and recognize how demoralized teachers are at the moment. Almost three-fourths of teachers indicated that they considered leaving their job at the end of the 2021–2022 school year, a figure that was up sharply from the year prior [20], which was much greater than historical trends [1,64]. School leaders must take steps to provide appropriate resources to teachers and find ways to appropriately compensate teachers for the jobs they are completing [38]. Teachers have put in longer than usual hours throughout the pandemic [22] with little to no compensation. School leaders and policymakers need to compensate teachers appropriately for their work and will continue to put into educating the future—especially if they want to ensure that they remain in the classroom beyond the pandemic.

*5.2. Limitations and Future Directions*

The current study is one of the first to explore teachers' experiences of morale and mental health at the end of the 2021–2022 school year. The results provide a starting point for school leaders to support teachers and limit teacher attrition as schools return to post pandemic practices. As research continues in these critical areas, it is important to note some of the current study's limitations that others may look to address in future studies. First, though the current study had a larger number of teachers, the sample lacked diversity. Researchers should work to have more robust and diverse samples that may glean further insights into teachers' experiences of morale and mental health. Second, data collection occurred at one point during the school year. Future research would benefit from a longitudinal design that examines changes in teacher burnout throughout the school year.

Additionally, as research has indicated throughout the pandemic, there is a concern about teachers' overall well-being. Future research should consider specific interventions to support teacher well-being. This will require more advanced research designs including different approaches to exploring teachers' experiences. The sampling procedure also represents a limitation. The anonymous link to the survey was posted in several places across multiple social media platforms to reach a large number of teachers. However, it should be acknowledged that teachers who elected to participate in this research and complete the survey may differ in important ways from those who never encountered the social media posts or did so and elected not to participate. Finally, as we discussed previously, the current study used a limited approach to measuring several constructs including teacher morale and mental health. This was aimed at limiting the burden placed on teachers needed to dedicate to participating in the study. Future studies should consider more robust teacher morale and mental health measures and explore other predictor variables that may decrease or increase teacher well-being.

In conclusion, the current study was one of the first studies to focus on teachers' experiences of morale and mental health as schools looked to return to normalcy at the end of the COVID-19 pandemic. Specifically, the current study sheds light on variables that may predict teacher morale, mental health, and job satisfaction. These results are important as states and schools continue to see an exodus of teachers leaving the profession. Moving forward, school leaders can use the results from the current study as a starting point to support teachers.

**Author Contributions:** Conceptualization, T.P. and D.T.M.; methodology, T.P. and D.T.M.; software, D.T.M.; validation, T.P. and D.T.M.; formal analysis, D.T.M.; investigation, T.P., D.T.M., S.M.L. and N.M.N.; resources, T.P., D.T.M., S.M.L. and N.M.N.; writing—original draft preparation, T.P., D.T.M., S.M.L. and N.M.N.; writing—review and editing, T.P., D.T.M., S.M.L. and N.M.N.; project administration, T.P. and D.T.M. All authors have read and agreed to the published version of the manuscript.

**Funding:** This research received no external funding.

**Institutional Review Board Statement:** This study received IRB approval from Auburn University (22-210 EX 2204) and Christopher Newport University (1906155-1).

**Informed Consent Statement:** Informed consent was obtained from all subjects involved in the study.

**Data Availability Statement:** Please contact first author for information about data availability.

**Conflicts of Interest:** The authors declare no conflict of interest.

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
