# Peer review of "Teacher Morale and Mental Health at the Conclusion of the COVID-19 Pandemic"

_education, doi:10.3390/educsci13121222_

Round 1

Reviewer 1 Report

Comments and Suggestions for Authors

the artical is intersting and importent. 

 It is necessary to clarify the topic of Teacher Morale. New articles from recent years should be added. What are the meanings? Why is it important to examine this (in light of the corona epidemic) in the teaching profession?

The research tools should be detailed. What processes were followed to perform validation and reliability?

n the summary and conclusions, it is worth emphasizing the importance of the findings and their implications for education systems. It is worth adding operative recommendations to change and improve the moral-mental state of teachers in the post-Covid era.

The research is of primary and important interest.

Author Response

Dear Reviewer,

Thank you for the quick and positive feedback on the article submitted to Education Sciences. We also appreciate the opportunity to revise and resubmit our article. To address all areas of concern, we have used the reviewers’ comments to guide the edits. To facilitate your review, we have highlighted all edits in the article document. Below, you will find Editor/Reviewer comments verbatim in black font; our responses are in red.

Thank you again for considering this article for a future issue of Education Sciences. If you have questions about the submission or need further information, please do not hesitate to let us know.

Reviewer 1

the artical is intersting and importent.

 It is necessary to clarify the topic of Teacher Morale. New articles from recent years should be added. What are the meanings? Why is it important to examine this (in light of the corona epidemic) in the teaching profession?

We have added some more recent articles to this section and added a summary sentence to state the purpose of examining teacher morale even beyond the COVID-19 pandemic.

The research tools should be detailed. What processes were followed to perform validation and reliability?

We have provided more detail about the measures in the method section.

In the summary and conclusions, it is worth emphasizing the importance of the findings and their implications for education systems. It is worth adding operative recommendations to change and improve the moral-mental state of teachers in the post-Covid era.

We have added more specific implications for school leaders to support teacher morale and mental health to the implications section.

The research is of primary and important interest.

Reviewer 2 Report

Comments and Suggestions for Authors

Excellent work on two important aspects of Teacher well-being in the post-pandemic era with useful implications for school leadership within the US educational context.

Please see more comments in the attached file.

Comments on the Quality of English Language

Very minor corrections are pending.

Author Response

Dear Reviewer,

Thank you for the quick and positive feedback on the article submitted to Education Sciences. We also appreciate the opportunity to revise and resubmit our article. To address all areas of concern, we have used the reviewers’ comments to guide the edits. To facilitate your review, we have highlighted all edits in the article document. Below, you will find Editor/Reviewer comments verbatim in black font; our responses are in red.

Thank you again for considering this article for a future issue of Education Sciences. If you have questions about the submission or need further information, please do not hesitate to let us know.

Reviewer 2

  1. TITLE AND ABSTRACT

The title effectively encapsulates the article's subject matter, and the abstract provides an overview of the research focus, its relevance to the field of education as well as information on the sample size and the statistical methods used for data analysis.

Suggestions:

Title: Teacher Morale and Mental Health at the End of the COVID-19 Pandemic

We appreciate this suggestion, but conclusion and end have similar meaning, thus, we have decided to keep Conclusion in the title.

Abstract: Key results concerning the predictor variables influencing TMH can also be mentioned here as they constitute the focal point for the adoption of initiatives followed by school leaders in future similar events.

Due to word limitations for the abstract we did not add the significant predictors to the abstract.

  1. INTRODUCTION

I guess the section ‘Teacher Morale and Mental Health at the Conclusion of the COVID-19 Pandemic’ is the Introduction – if so, say it.

It effectively summarizes the current situation on TMH throughout the pandemic period and sets the scene for the study by succinctly stating the respective RQs.

We have updated this header to say Introduction.

  1. LITERATURE REVIEW

The section ‘Theoretical framework’ can be part of the literature review section (in fact no theoretical framework is specifically mentioned there) as relevant literature on teacher morale and mental health is key variables of the variable is reviewed. Then, authors can explain their interest in studying these two aspects of teacher wellbeing by referring to the demoralization issues teachers have faced throughout the COVIF-19 study and how this situation may have changed in the post-pandemic years.

We have updated the headings used in this section.

  1. METHODOLOGY

Sound information is offered in the sections ‘Procedures and Sampling’ and ‘ Participants’. However, with respect to the instrument used for data collection, information is somewhat missing. The survey consists a lot of scales measuring a lot of different things that indirectly measures TMH and morale. Specify what these dimensions are and state the number of items per dimension. Also, validity and reliability ratings are only offered for some but not for all scales of the survey. ‘Data analysis’ section is fine.

We have provided more information about measures used in the method section.

  1. RESULTS

Results of correlation and hierarchical regression models are properly cited and reported.

  1. DISCUSSION AND CONCLUSIONS

Key findings are discussed and properly lead to school leadership suggestions to boost teachers’ morale. Do these solutions meant to be followed by school leaders to promote TMH and morale only after times of crisis, like COVID or, in other cases too (e.g. to reduce teacher attrition rate). I assume both. Specify.

We have specified that is important for both in the implications section.

Overall, the study can be considered a useful contribution to the field of teacher wellbeing in the post-COVID-19 era by measuring teacher mental health and morale adding up the literature on this topic in times of crisis. However, validity of the data is jeopardized as there is no accurate description of all dimensions used to measure the two variables under consideration. It is rather unclear which scale measures what exactly, although the authors make the excuse that they used truncated scales to shorten the version of the survey they were using.

The discussion is consistent with the evidence leading to implications for leadership on the school level with regard to remedial actions that could be assumed to improve TWB. The selection of references is appropriate. Unfortunately, no tables or figures were included in the manuscript I received, so no comments can be made in this respect.

We have provided more information about the measures included in the instrumentation in the method section.
